# Genome-Wide Identification of the *Hypericum perforatum* WRKY Gene Family Implicates *HpWRKY85* in Drought Resistance

**DOI:** 10.3390/ijms24010352

**Published:** 2022-12-26

**Authors:** Wen Zhou, Shu Yang, Lei Yang, Ruyi Xiao, Shiyi Chen, Donghao Wang, Shiqiang Wang, Zhezhi Wang

**Affiliations:** 1Key Laboratory of the Ministry of Education for Medicinal Resources and Natural Pharmaceutical Chemistry, National Engineering Laboratory for Resource Development of Endangered Crude Drugs in Northwest of China, Shaanxi Normal University, Xi’an 710062, China; 2Shaanxi Engineering Research Centre for Conservation and Utilization of Botanical Resources, Xi’an Botanical Garden of Shaanxi Province, Institute of Botany of Shaanxi Province, Xi’an 710061, China

**Keywords:** *Hypericum perforatum*, WRKY gene family, expression patterns, drought resistance

## Abstract

WRKY, named for its special heptapeptide conserved sequence WRKYGOK, is one of the largest transcription factor families in plants and is widely involved in plant responses to biotic, abiotic, and hormonal stresses, especially the important regulatory function in response to drought stress. However, there is no complete comprehensive analysis of this family in *H. perforatum*, which is one of the most extensively studied plants and is probably the best-known herbal medicine on the market today, serving as an antidepressant, neuroprotective, an antineuralgic, and an antiviral. Here, we identified 86 *HpWRKY* genes according to the whole genome database of *H. perforatum*, and classified them into three groups through phylogenetic analysis. Gene structure, conserved domain, motif, cis-elements, gene ontology, and expression profiling were performed. Furthermore, it was found that *HpWRKY85*, a homologous gene of *AtWRKY75*, showed obvious responses to drought treatment. Subcellular localization analysis indicated that this protein was localized in the nucleus by the Arabidopsis protoplasts transient transfection. Meanwhile, *HpWRKY85*-overexpressing Arabidopsis plants showed a stronger ability of root growth and scavenging endogenous reactive oxygen species. The results provide a reference for further understanding the role of *HpWRKY85* in the molecular mechanism of drought resistance of *H. perforatum*.

## 1. Introduction

*Hypericum perforatum*, also known as St. John’s wort, has a history of being used as a traditional herbal medicine in Eurasian nations [1,2]. It is also one of the traditional Chinese herbal medicines in China, first collected in *the Supplement to Compendium of Materia Medica* (Ben-Cao-Gang-Mu-Shi-Yi) [3]. *H. perforatum* has a long history in western countries in the treatment of irritability, depression, anxiety, poor concentration, and mental tension [4,5]. In recent years, it has been widely used for its the pharmacological effects; the commercial value of *H. perforatum* is becoming increasingly prominent and the market demand is increasing [6]. The collection of wild resources and the existing artificial planting scale are struggling to meet the market demand. It is necessary to expand the planting area of *H. perforatum*. This species has strong adaptability to the environment and can be popularized in most parts of the country. However, the humid and semi-humid areas are the main production areas of food crops and commercial crops in China, and the land area available for planting *H. perforatum* and other medicinal plants is limited. Therefore, promoting the planting of *H. perforatum* in arid and semi-arid areas has become one of the important means of alleviating its resource shortage. Understanding the drought resistance molecular mechanism of *H. perforatum* is foundational to realizing its large-scale planting in arid and semi-arid areas.

Transcription factors often play a vital role in the regulatory mechanism in gene expression. The WRKY transcription factor is one of the most important transcription regulators in plants, found in sweet potato for the first time [7]. As more researchers have participated in this field, more WRKY proteins have been demonstrated in plants. WRKY transcription factors are usually a sequence approximately by 60 amino acids in length containing the WR(K)KYGQ(K/N/E)K sequence and C2HH(C) zinc finger structure [8,9]. The function of WRKY transcription factors to regulate the expression of target genes needs to meet the following two conditions. Firstly, the WRKYGQK sequence successfully recognizes and binds to the (T)(T)TGAC(C/T) sequence located in the promoter region called the W-box. Secondly, Zn^2+^ was successfully captured by the zinc finger structure [10]. Currently, the most widely used WRKY transcription factor classification system was established with reference to the Arabidopsis genome in 2000 [11].

The WRKY gene family regulates numerous biological processes, including seed dormancy and germination [12], stem elongation [13], embryogenesis [14], epidermal hair and seed coat development [15], trichome development, injury, senescence [16], pathogen resistance [17], regulation of flowering time [18], fruit ripening [19], secondary metabolites biosynthesis [20,21], biotic, and abiotic stresses [22], etc. WRKY transcription factors are also involved in abscisic acid (ABA), ethylene (ETH), jasmonic acid (JA), salicylic acid (SA), and MAPK-mediated signal transduction processes [23]. For instance, *AtWRKY8* confers TMC-cg resistance in Arabidopsis through regulation of ABA and ETH signaling pathways [24]. *AtWRKY70* is a key node in the salicylic acid and jasmonic acid resistance pathways, and its loss of function confers susceptibility to powdery mildew, botrytis and leaf spot in Arabidopsis [25]. Additionally, WRKY transcription factors play crucial roles in the regulation of plant stress resistance, including drought, soil salinity, cold, high temperature, nutrient deficiency, heavy metal toxicity, ultraviolet light, and mechanical damage [26]. For example, wheat *TaWRKY10* and *TaWRKY44* can enhance the drought resistance of transgenic tobacco by regulating the accumulation and elimination of ROS and the expression of stress functional genes [27,28]. Rice *OsWRKY11* is regulated by the *HSP101* promoter, and its overexpression can enhance the heat tolerance of rice plants [29]. Overexpression of maize *ZmWRKY33* in Arabidopsis can significantly enhance the low temperature stress resistance of Arabidopsis [30]. Overexpression of cotton *GhWRKY25* and *GhWRKY39-1* in tobacco enhanced its tolerance to salt stress in plants [31,32].

In this study, we identified 86 *HpWRKY* genes in *H. perforatum* and classified them into three groups. A comprehensive bioinformatics analysis, including the determination of phylogenetic analysis, gene structure, conserved domain, motif, cis-elements, and gene ontology, were performed. Furthermore, RNA-seq analysis was performed to understand the gene expression pattern of WRKY gene family members in different tissues under drought stress conditions in *H. perforatum*. Quantitative real-time PCR (qRT-PCR) was utilized to validate that the expression profiling data of *HpWRKY* genes exhibit different expression patterns. A subcellular localization experiment was conducted to explore the expression position of *HpWRKY* transcription factors. Finally, we functionally characterized a transcription factor *HpWRKY85* in response to drought stress, and further explored the phenotypic and physiological characteristics of overexpressed transgenic Arabidopsis plants in response to drought treatment so as to provide data for further understanding the role of *HpWRKY85* in the molecular mechanism of drought resistance of *H. perforatum*.

## 2. Results

### 2.1. Identification and Phylogenetic Analysis

The HMM of the WRKY domain (PF03106) was employed to search against the *H. perforatum* genomes database and a total of 86 *HpWRKY* genes named as *HpWRKY1*-*HpWRKY86* were identified. Pfam and SMART analyses showed that these selected proteins contained a complete WRKY domain. The basic physical and chemical properties were shown in Appendix A, in which the length of *HpWRKY* gene ranges from 519 bp (*HpWRKY81*) to 6194 bp (*HpWRKY76*). The length of *HpWRKY* cDNA and protein ranges from 339 bp and 112 aa (*HpWRKY9*) to 2172 bp and 723 aa (*HpWRKY71*). The predicted MW of the proteins ranged from 12.41 kDa (*HpWRKY9*) to 78.51kDa (*HpWRKY71*), and the PI ranged from 4.74 (*HpWRKY60*) to 10.03 (*HpWRKY65*). Subcellular localization prediction showed that except *HpWRKY16* was expressed in both the nucleus and cytoplasm; the other 85 WRKY were all located in nucleus (Appendix A).

As shown in Figure 1, a neighbor-joining phylogenetic analysis of 86 *H. perforatum* and 72 Arabidopsis WRKYs sequences using default parameters and 1000 bootstraps. According to the phylogenetic analysis, 86 HpWRKY proteins could be classified into three groups, of which 23 HpWRKY proteins belonged to Group 1, 55 HpWRKYs belonged to Group 2, and 8 HpWRKYs belonged to Group 3. Group 2 was further divided into five subgroups (2a–2e), containing 6, 9, 22, 5, and 13 proteins respectively. The classification of HpWRKYs confirmed the diversity of their protein structures, suggesting that different subfamily members may have different regulatory functions, and that orthologous genes are likely to have similar functions. For instance, *AtWRKY72* utilizes an SA-independent defense mechanism and plays a role in the basic defense against various pathogens and pests [33]. Then, as a homologous gene of *AtWRKY72*, *AtWRKY18* may have the same function. *AtWRKY75* not only enhances drought and salt tolerances; it isconsidered to be a positive regulator in the JA or SA mediated defense signal response to the necrotrophic fungal pathogens, which can cause soft rot in a wide range of plant hosts [34]. In other words, *HpWRKY54*, *HpWRKY85*, and *HpWRKY74* may have the same regulation mechanism.

### 2.2. Sequence Analysis

According to the multiple sequence alignment of HpWRKY protein core domain (Appendix A), the zinc finger structure of WRKY protein of Group 1 was C-X4-C-X22-23-H-X1-H type. Except for HpWRKY16, the remaining 22 proteins in Group 1 contain two WRKY domains, of which the two conserved domains of 19 members are complete. The C-terminal of HpWRKY31 lacks the zinc finger structure, and the N-terminal zinc finger structure of HpWRKY32 and HpWRKY50 is incomplete. In Group 2, 55 members contained only one conserved domain, of which 52 HpWRKYs carried the conserved WRKYGK sequence, while HpWRKY9, HpWRKY56, and HpWRKY83 showed the WRKYGKK sequence. C2H2 zinc finger structure was observed in 52 HpWRKYs (C-X4-5-C-X23-H-X1-H). Group 3 contains eight HpWRKY proteins, seven of which contain WRKYGQK sequence, and the zinc finger structure is C-X7-C-X24-H-X 1-C type.

The gene structure analysis of *H. perforatum* WRKY is shown in Figure 2A. Except for *HpWRKY81* which has no intron, the number of introns in other genes ranges from 1 to 5. Among them, eight members have only one intron, and up to 77 (91%) HpWRKYs possessed 2–5 introns. Further analysis showed that the introns of *HpWRKY* had three phases: 0, 1, and 2. The phase of the first two introns of most sequences in Group 1 was 0 and the last two introns was 2. Except for *HpWRKY5*, the intron phase of all other members in Group 2a and Group 2b was 0, and the intron phase of all members in Group 3 was 2. It can be seen that the structure of *HpWRKY* is conservative in the same group.

The results of motif analysis by MEME were shown in Figure 2B. Motif 1, 2, and 16 were all seven peptide sequences of WRKYGQK (Appendix A). Except that motif 16 existed only in *HpWRKY31* and *HpWRKY81* with zinc finger deficiency, motif 1 and motif 2 were widely distributed in other protein sequences. HpWRKY members in the same group have similar motif composition. For example, motif 8 and 9 were found only in Group 2a and Group 2b, motif 7, motif 11 was found only in Group 1, and motif 10 was found only in the Group 2c members. Overall, the conserved motif composition and similar gene structure of HpWRKY members in the same group further verified the classification and phylogenetic relationship.

### 2.3. Cis-Acting Elements Analysis and GO Annotation

The cis-acting elements in the 1.5 kb upstream promoter region of all *HpWRKY* genes were predicted and analyzed by PlantCARE, as shown in Appendix A. Almost all cis-acting elements were related to hormone and biological stress response, including salicylic acid (SA), auxin (IAA), abscisic acid (ABA), gibberellin (GA), methyl jasmonate (MeJA), and fungi. Among them, SA response elements were found in the promoter region of 60 *HpWRKY* genes, accounting for the largest proportion. ABA and MeJA (ABRE and CGTCA/TGACG-motif) response elements were found in the promoter regions of 47 and 54 genes, respectively. In addition, some elements related to abiotic stresses such as light, injury, low/high temperature, anaerobic induction, and drought were found in a large number of *HpWRKY* genes. TC rich repetitive elements (involved in defense and stress responsiveness) were found in the promoter region of 54 genes, and MBS (drought inducible) elements appeared in the promoter region of 63 genes, indicating that most *HpWRKY* genes were involved in drought stress response. A total of 51 *HpWRKY* genes contained WRKY binding sites (W-box), indicating that these genes may be self-regulated or cross regulated with other genes.

Based on amino acid similarity, 86 HpWRKY proteins were classified by Blast2GO. It can be seen from Appendix A that the proteins were annotated to 21 subclasses of biological processes, molecular functions, and cell components. All HpWRKY proteins were annotated in the functional subclasses of cell part, cell, organelle, binding, transcription regulator activity, metabolic process and cellular process.

### 2.4. Transcript Abundance Profiling

The formula log_10_(FPKM) from the RNA-seq data (flowers, leaves, roots, and stems) referred to in the previous description [35] were applied to hierarchical clustering (Figure 3A). Eight of eight-six *HpWRKY* genes were not examined in the library, possibly because they were pseudogenes, or the expression was extremely low and could be ignored. *HpWRKY57* showed the highest transcript accumulation in all tissues compared with the expression levels of *HpWRKY22*, *HpWRKY42*, and *HpWRKY61*. According to the phylogenetic tree analysis, *HpWRKY57* and *AtWRKY40* belong to the same subfamily. Therefore, *HpWRKY57* may also have similar functions in plant drought stress responses. Some WRKY genes preferentially accumulate transcripts in one tissue; for example, *HpWRKY35*, *HpWRKY84*, and *HpWRKY56* showed the highest expression levels in leaves. *HpWRKY35* is an ortholog of *AtWRKY70*, a negative regulator of developmental senescence, so it may also be an inhibitor in the process of transcription regulation. Through the verification of qRT-PCR between different organs of the six *HpWRKY* genes (Appendix A), it was found that the expression patterns were consistent with the heat map, which proved the reliability of the transcriptome data. *HpWRKY4* and *HpWRKY44* were mainly expressed in roots, *HpWRKY35* was mainly expressed in leaves, and *HpWRKY85* shows a high expression in both roots and leaves.

Ten *HpWRKY* genes (Figure 3B) distributed in different subgroups were selected to investigate the expression patterns under SA and GA hormone treatments and drought stress at five time points (0, 1, 3, 6, and 12 h). It was obvious that hormone treatment and drought stress can induce these genes to produce different levels of response. Almost all ten genes showed up-regulation under SA, MeJA, and drought treatments. Drought treatment up-regulated *HpWRKY1* expression, while MeJA and SA treatment down-regulated *HpWRKY1* expression. For SA treatment, the expression levels of *HpWRKY7, HpWRKY27*, *HpWRKY35*, and *HpWRKY79* increased gradually and reached the peak at 3 h, while the expression levels of *HpWRKY42*, and *HpWRKY85* accumulated rapidly at 1 h and then began to decline. For MeJA treatment, the expression levels of *HpWRKY8*, *HpWRKY27*, *HpWRKY35*, *HpWRKY42*, *HpWRKY79*, *HpWRKY83* and *HpWRKY85* reached the peak at 12 h. For drought treatment, the expression of *HpWRKY42* and *HpWRKY61* reached the peak at 1 h. Significantly, the expression of *HpWRKY85* was increased the most, which was 85, 58, 63, and 188 times higher at 0, 1, 3, 6, and 12 h than the control, respectively.

### 2.5. Characterization of Transcription Activity of HpWRKY85

According to the above phylogenetic and expression profile analysis, *HpWRKY85*, as a homologous gene of *AtWRKY75*, showed obvious response to drought treatment. The cDNA of the *HpWRKY85* has a total length of 567 bp and encodes 188 amino acids. The MW of the protein is 21,133.60 Da and the PI is 9.49. According to the prediction of bioinformatics analysis, HpWRKY85 is an intracellular protein without a signal peptide and transmembrane domain. It has strong hydrophilicity and is located in the nucleus. Subsequently, the CDS of *HpWRKY85* was constructed on the expression vector PBI221-GFP. By transforming Arabidopsis protoplast and observing the fluorescence information, it was found that the GFP green fluorescence of the empty vector EV-GFP existed in the nucleus and cytoplasm, while the fluorescent signal of *HpWRKY85*-GFP was exclusively expressed in the nucleus, which was consistent with the prediction of bioinformatics (Figure 4A and Appendix A). Since WRKY transcript factors are known to bind to the W-box sequence (C/T)TGAC(T/C), our yeast one-hybrid test demonstrated that *HpWRKY85* was bound to W-box with a conservative sequence (Figure 4B). In conclusion, the above results showed that *HpWRKY85* is a typical WRKY transcription factor.

### 2.6. Overexpression of HpWRKY85 in Arabidopsis

Arabidopsis WT lines was transfected with pEarleyGate202-*HpWRKY85* transformation solution to study the gene function of *HpWRKY85*. The seeds of Arabidopsis were collected by flower soaking method and sown on the screening medium to obtain candidate transgenic plants. The positive plants could grow normally on the medium containing antibiotics (Figure 5A). The target gene was amplified and tested to further verify whether the transgenic plants were positive. As shown in Figure 5B, *HpWRKY85* were cloned from ten candidate transgenic lines, but no bands were found at 500–600 bp in WT. In order to obtain the lines with the largest up-regulation at the transcriptional level, three transgenic lines OE1-*HpWRKY85*, OE2-*HpWRKY85*, and OE3-*HpWRKY85,* were finally selected by semi quantitative RT-PCR for the next drought resistance function analysis (Figure 5C). As shown in Figure 5D, when OE and WT plants grow on one half-strength MS medium, there is no significant difference in root length. With the drought treatment simulated by D-mannitol, the growth of plant roots was inhibited. At the same time, the main root length and the number of lateral roots of the OE lines was about 1.45 and 1.56 times that of WT (Figure 5E).

In order to figure out whether *HpWRKY85* can enhance the drought of plants, four-week-old Arabidopsis seedlings were used for physiological determination. The growth of OE lines was obviously better than that of WT after without watering for 8 days (Figure 6A), and the MDA and H_2_O_2_ contents in OE lines were significantly lower than in WT (Figure 6B). From the CM-H_2_DCFDA staining, it can be seen that the fluorescence intensity in OE leaves (Figure 6C) is lower than that in WT under drought treatment. These results showed that the content of ROS in transgenic leaves was lower than that in WT.

## 3. Discussion

In recent years, with the continuous understanding of the pharmacological effects of the bioactive substances of *H. perforatum*, it has become one of the best-selling medicinal plants in the world. Since the main active component of *H. perforatum* is mainly distributed in the above-ground tissues such as flowers and leaves, its harvesting period is concentrated in the flowering period. This results in the interruption of flowering and fruiting, which in turn affects the reproduction of individual plants. Coupled with long-term development and utilization, wild resources gradually become scarce, and artificial cultivation often has problems such as unstable yield and effective component content. Therefore, the existing resources struggle to meet the market demand, and it is imperative to expand the planting area. *H. perforatum* does not have strict habitat requirements and can also grow in arid and semi-arid areas. Popularizing the planting of *H. perforatum* in this area can not only meet the current demand for expanding the planting area, but also improve the land utilization rate of our country. Understanding the effects of drought stress on plants will help to clarify the molecular mechanism of drought resistance of *H. perforatum*, and then provide a theoretical basis for its large-scale planting in arid and semi-arid regions. WRKY, as a key transcription factor in plants, plays an indispensable role in regulating plant growth and development, stress response, and other physiological processes. After sweet potato SPF1 was reported as the first WRKY transcription factor in 1994 [7], a large number of WRKY family members were found and identified in other species. With the development of biological information technology, transcriptome and genomic data of many species have been published. 45, 72, and 85 WRKY family members were identified in Hordeum vulgare [36], Arabidopsis thaliana [37], and Salix suchowensis [38], respectively. Members of WRKY gene family in vegetables, fruits, commercial crops, and medicinal plants have been found in succession. For example, there are 71, 59, 82, and 63 WRKY transcription factors in pepper Capsicum annuum [39], grape Vitis vinifera [40], potato Solanum tuberosum [41], and Dendrobium officinale [42], respectively.

Based on the genomic database of *H. perforatum* in our lab, the 86 WRKY of *H. perforatum* were characterized and divided into three groups by phylogenetic analysis, of which the second group had the largest number of members. Studies have shown that in many plant WRKY transcription factor classifications, Group 2 members also account for the largest proportion, such as radish [43] and pepper [39]. The second group is divided into five subgroups, in which the members of Groups 2D and 2E are concentrated in two branches of the same branch, and the same is true of Groups 2a and 2B. This indicates that they are more closely related in the evolution of WRKY gene, which is consistent with the evolutionary research results of other researchers on wheat [44] and cassava [45]. By analyzing the core domain of WRKY protein of *H. perforatum*, it is found that the conservative domains in the same group are basically the same, which implies that there are similar functions among WRKY members in the same group, and there are specific conservative domain compositions among different groups [46]. The analysis of cis-acting elements of promoter showed that *HpWRKY* was involved in the transcriptional regulation of plant defense and stress response. The expression pattern of *HpWRKY* showed a diversified trend, and the expression in tissues was specific. This indicates that these *HpWRKY* members may participate in different physiological and biochemical processes. Similar results also exist in cucumber [47], cotton [48], and Arabidopsis [11]. Through hormone induction and drought stress treatment, it was found that the candidate genes were up-regulated in varying degrees, among which the expression of *HpWRKY85* was up-regulated the most, indicating that this gene product may be the key to participation in the signal transduction pathway.

The results of simulated drought treated with D-mannitol showed that the OE-*HpWRKY85* Arabidopsis had longer root length and better growth than the WT. Studies have shown that the length of the root system is an important index to evaluate drought resistance [49]. The longer the root length, and the deeper the rooting depth, the more helpful it is to enhance the drought resistance of the plant [50]. Therefore, the above results indicate that the *HpWRKY85* gene has a certain relationship with drought resistance. Under drought conditions, the lower the content of ROS in plants, the stronger the drought resistance of plants. This study showed that under drought stress, compared with WT Arabidopsis, the ROS accumulation of OE lines is less, which is consistent with the research results of Wang et al.’s study on tobacco *TaWRKY10* [27]. MDA is a product of membrane lipid peroxidation, which can be used as a key indicator to judge membrane damage. This study found that under drought stress, compared with WT, the content of MDA in the OE lines was less. The above results indicate that overexpression of *HpWRKY85* can reduce the degree of membrane lipid peroxidation in plants so that plants have strong drought resistance. In conclusion, through the systematic analysis of the WRKY family members of *H. perforatum*, this study preliminarily confirmed that the gene *HpWRKY85* can reduce the degree of membrane lipid peroxidation, reduce the accumulation of reactive oxygen species in plants, reduce the damage of stress to plants, and play a positive role in the process of plant drought resistance.

## 4. Materials and Methods

### 4.1. Identification and Phylogenetic Analysis

The amino acids sequences of the 72 Arabidopsis WRKY proteins were downloaded from the Arabidopsis Information Resource Database (TAIR, http://www.Arabidopsis.org/ (accessed on 20 August 2022) and the hidden Markov model of the WRKY domain (PF03106, http://pfam.sanger.ac.uk/ (accessed on 20 August 2022) was used as a query to search against the *H. perforatum* genome assembly (BioProject PRJNA588586; BioSample SAMN13254339) of our lab [10,33]. The raw data of the RNA-seq analysis used in this study was submitted to the Sequence Read Archive (SRA) at NCBI under accession numbers SRR8438983-SRR8438986. Only those which have complete WRKY domains were analyzed using Pfam and SMART analyses as members of the *H. perforatum* WRKY gene family. A neighbor-joining (NJ) phylogenetic and evolutionary tree was constructed using MEGA 11, which included the amino acid sequences of 158 WRKY proteins (72 WRKYs in Arabidopsis and 86 WRKYs in *H. perforatum*).

### 4.2. Sequence Analysis

ExPASy https://web.expasy.org/protparam/ (accessed on 20 August 2022) was used to obtain the sequence length, molecular weight, isoelectric point, and subcellular location prediction of the candidate HpWRKY proteins [51]. DNAMAN7 was used to align multiple sequences of WRKY core domain sequences [52]. Gene Structure Display Server (http://gsds.cbi.pku.edu.cn accessed on 20 August 2022) was used to predict the *HpWRKY* genes intron–exon structures by using the genomic sequences and the corresponding coding sequences (CDS). The conservation of the HpWRKY protein motif was analyzed by MEME online program https://meme-suite.org/meme/tools/meme (accessed on 20 August 2022) with the maximum number of motifs set to 16 and optimum width of motif from 6–200. The motif sequences were manually adjusted using Weblogo http://weblog.berkeley.edu/logo.cgi (accessed on 20 August 2022) to display the characteristics of the WRKY domains and, again using Weblogo http://weblog.berkeley.edu/logo.cgi (accessed on 20 August 2022), to adjust the sequence to show the characteristics of the WRKY domain [53]. The promoter region (1.5 kb upstream of genomic DNA sequences) of the *HpWRKY* genes was uploaded to PlantCARE http://bioinformatics.psb.ugent.be/webtools/plantcare/html/ (accessed on 20 August 2022) to identify putative cis-elements. Blast2GO software is used to obtain GO annotations for each HpWRKY and to perform functional classification on 86 HpWRKY proteins [54]. Then, Wego software was used to run GO function classification and distribution at the macro level. The subcellular localization of HpWRKY was predicted on CELLO http://cello.life.nctu.edu.tw/ (accessed on 20 August 2022) by amino acid sequence.

### 4.3. Plant Materials and Stress Treatments

The seeds of *H. perforatum* were collected from Qinling Mountains (Liuba County, Shaanxi Province), disinfected with 12% chloro solution for 8 min, and washed with sterile water 6 times. Then, the seedlings were sown on MS solid medium and cultured in an artificial climate incubator (23 ± 2 °C, 16 h light/8 h dark). Tissue samples (stem, root, leaf, and flower) were collected from 2-year-old plants to analyze the tissue-specific expression profile of *HpWRKY* genes. The 2-month-old seedlings were used for stress related expression profile analysis. For the drought treatment, the seedlings were treated with 200 mmol/L D-mannitol solution. For the hormonal treatment, the aseptic seedlings cultured in the incubator were subjected to 10 mM salicylic acid (SA) and 200 mM methyljasmonate (MeJA) in MS liquid medium. All samples were collected at 0 h, 1 h, 3 h, 6 h, and 12 h, immediately frozen in liquid nitrogen and stored at −80 °C refrigerator for subsequent use. The transcription profile of *HpWRKY* genes was calculated as 1000-base exon model fragment per million map reads (FPKM).

Arabidopsis wild-type (WT) Col-0 seeds were kept in our laboratory. For the seed sterilization method and growth conditions, refer to the previous description [55]. For drought treatment, the one-week-old aseptic seedlings were transferred to the one half-strength MS medium containing 250 mmol/L D-mannitol for 15 days to observe their phenotypes. Four week-old field seedlings were not watered for 8 consecutive days for physiological determination.

### 4.4. Gene Cloning, Vector Construction, and Genetic Transformation

*HpWRKY85* were cloned from *H. perforatum* cDNA using gene-specific primers (Appendix A). For the subcellular localization vector construction, the amplified products were inserted into PBI221-GFP to produce the 35S::WRKY-GFP constructs by the restriction enzyme digestion. For the overexpression (OE) vector construction, the amplified products were recombined into the pDONR207 and pEarleyGate202 Gateway vector (Invitrogen, St. Louis, USA) using the BP and LR recombination reaction to generate pEarleyGate 202-*HpWRKY85*. Then, the final OE plasmid and empty vector pEarleyGate 202 were transformed into Agrobacterium tumefaciens GV3101 (Weidi, Shanghai), and Arabidopsis WT plants was transfected by the flower dipping method [56]. The positive plants were screened according to the resistance of the recombinant plasmid to BASTA.

### 4.5. PCR Analysis, Subcellular Localization Analysis, and Yeast One-Hybrid Assay

Total RNA and DNA were extracted using the Plant Total RNA/DNA Isolation Kit (Sangon, Shanghai). The synthesis of cDNA was performed by using the 1st Strand cDNA Synthesis Kit (Vazyme, Nanjing, China). The ~900 bp CaMV35S promoter sequence was amplified from the genomic DNA to detect whether the recombinant plasmid was integrated into the plant genome to screen the positive transgenic Arabidopsis. Reverse transcription PCR (RT-PCR) was used to investigate the expression pattern of *HpWRKY85* in Arabidopsis WT and OE plants. *AtACT3* (NM179953) gene fragment was used as an internal control. GenScript https://www.genscript.com/tools/ (accessed on 20 August 2022) was used to design the quantitative primers (Appendix A). As previously described [35], qRT-PCR was carried out on a Roche LightCycler 96 system and performed three times for every three biological samples, including the negative control without template. *HpACT2* (MK054303) was used as an internal reference gene with the 2^-ΔΔCt^ method [57].

The transient transformation of Arabidopsis protoplasts was performed according to Dr. Sheen’s operation manual [58]. For specific solution preparation and operation steps, refer to the previous instructions [59]. To analyze *HpWRKY85* binding to W-box motif, yeast one-hybrid assay was performed. The promoter sequence of Arabidopsis RD29A gene (AT5G52310), containing triple W-box cis-acting elements was cloned into the pHis2 vector as the reporter construct pHis2-W-box [60]. The CDS of *HpWRKY85* was cloned into the pGADT7 vector as the effector pGADT7-*HpWRKY85*. The two recombinant vectors described above were transformed into yeast Y187 strain and screened on SD/-Trp/-Leu plates at 28 °C for 3 days. The surviving colonies were then transferred to SD/-Trp/-Leu/-His medium containing 50 mM 3-AT to observe the interaction between *HpWRKY85* and the W-box elements.

### 4.6. Physiological Assays and Statistics

One-week-old WT and OE aseptic seedlings of Arabidopsis under drought stress were analyzed to observe their phenotypes. One-month-old WT and OE field plants were analyzed for drought resistance. Malondialdehyde (MDA) and H_2_O_2_ concentration was measured using the MDA/H_2_O_2_ content assay kit (Sangon, Shanghai, China) with three biological and three technical replicates. The reactive oxygen species (ROS) level was measured using CM-H_2_DCFDA fluorescent dye (Sigma-Aldrich, St. Louis, MO, USA) and then examined for fluorescence signals by Leica stereomicroscope (Leica, Wetzlar, Germany). ANOVA analysis was used for statistical analysis, and the probability value *p* < 0.05 was considered to be statistically significant.

## Figures and Tables

**Figure 1 ijms-24-00352-f001:**
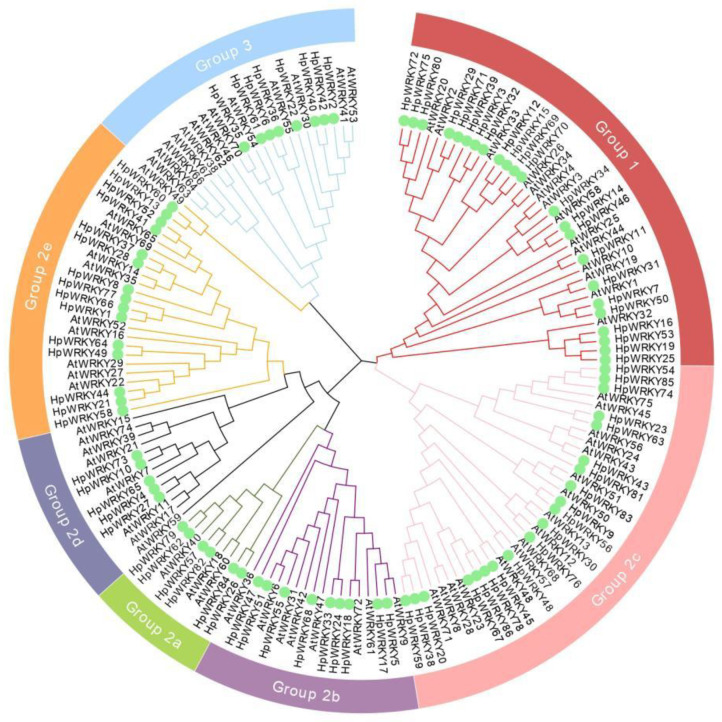
Phylogenetic analysis representing relationships between the WRKY proteins from *A. thaliana (At)* and *H. perforatum* (Hp). The neighbor-joining tree performed in MEGA 11.0 software including 72 WRKY proteins from Arabidopsis and 86 from *H. perforatum*. The green dots indicate the WRKY protein of *H. perforatum*. Different colors in the outermost circle represent different subfamilies.

**Figure 2 ijms-24-00352-f002:**
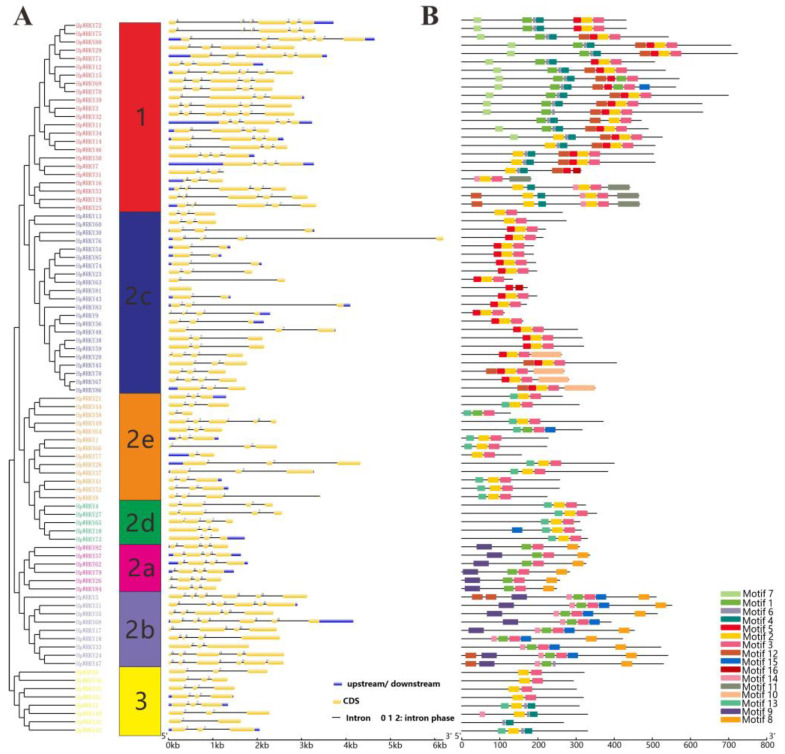
The phylogenetic relationships, gene structure, and composition of conserved motifs in WRKY from *H. perforatum*. (**A**) The phylogenetic on the left contains 86 WRKY proteins (named *HpWRKY1* to *HpWRKY86*). The yellow boxes, gray lines, and blue boxes in the exon-intron structure diagram represent exons, introns, and UTRs, respectively. (**B**) The motif patterns of 86 HpWRKY proteins.

**Figure 3 ijms-24-00352-f003:**
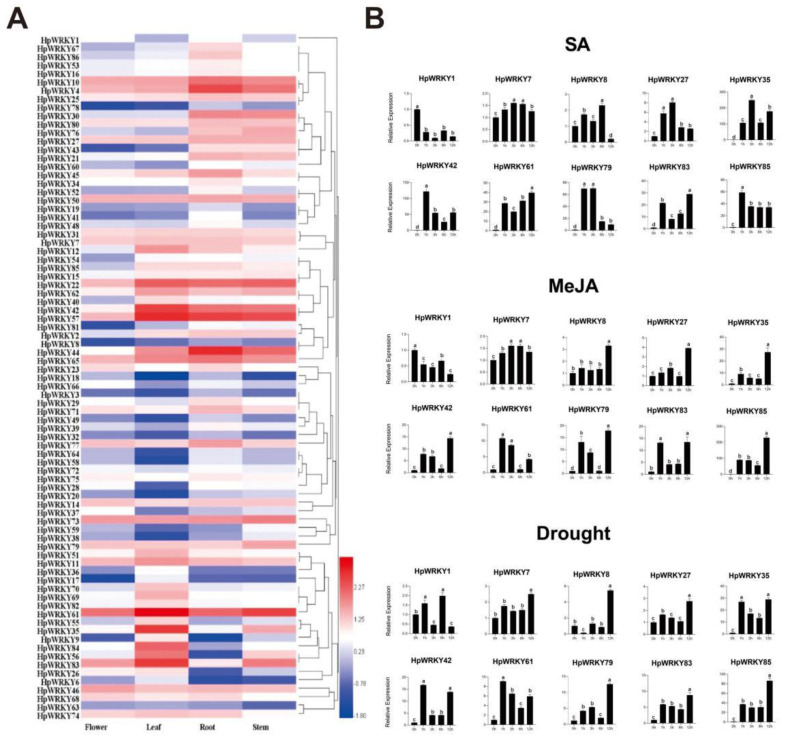
Expression profiles of *HpWRKY* genes. (**A**) Hierarchical clustering of expression profiles of *HpWRKY* genes in different tissues (root, stem, leaf, and flower; three biological replicates). log_10_(FPKM) values were displayed according to the color code (bottom left). (**B**) qRT-PCR assay-based expression patterns of 10 selected *HpWRKY* genes in 2-month-old *H. perforatum* leaves under SA, MeJA, and drought treatments. Data were normalized by *HpACT2* (MK054303), calculated with the equation 2^−ΔΔCt^. All data represent averages of three biological replicates. Error bars indicate ±SD. Different letters indicate significant differences from the control (*p* < 0.05) tested by one-way ANOVA.

**Figure 4 ijms-24-00352-f004:**
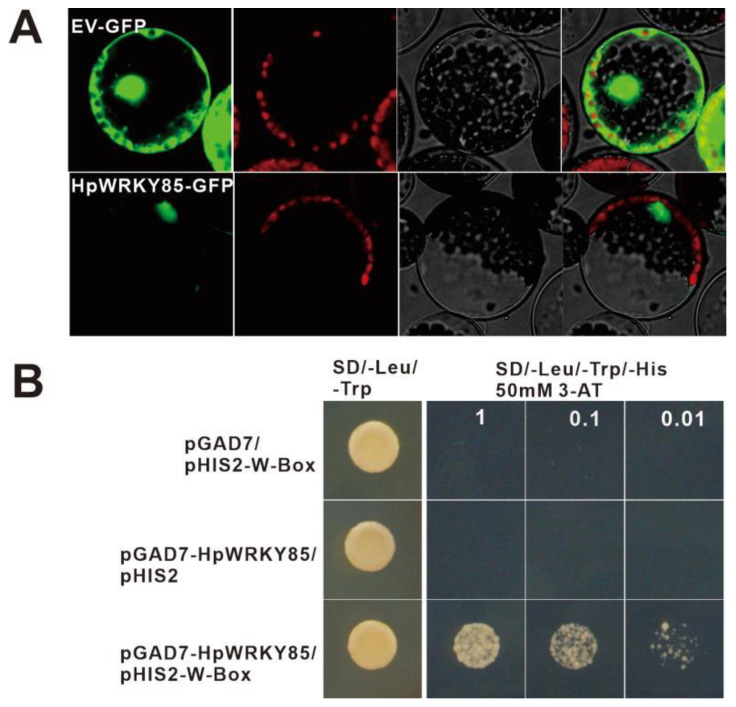
*HpWRKY85* is a typical WRKY transcription factor. (**A**) The subcellular location of HpWRKY85. From left to right are the subcellular localization of the target genes (GFP), chloroplast autofluorescence (Auto), bright-field (Bright), and Merged (Merge) images. Scale bars, 5 μm. (**B**) Yeast one-hybrid assay showed that *HpWRKY85* can bind to the W-box motif.

**Figure 5 ijms-24-00352-f005:**
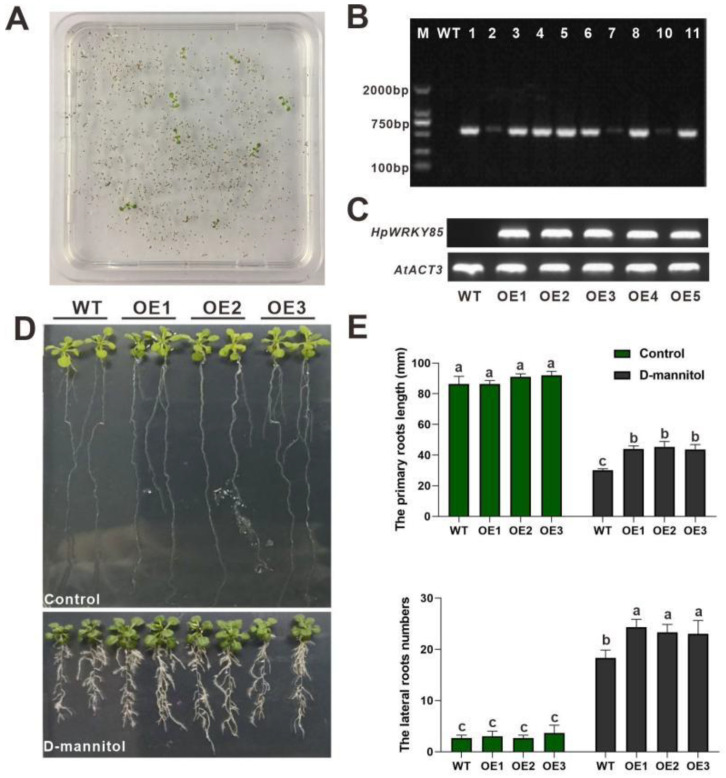
Function of *HpWRKY85* when expressed in Arabidopsis plants. (**A**) Screening of Arabidopsis transgenic plants, positive plants can grow normally on the screening medium. (**B**) PCR results of *HpWRKY85*. *HpWRKY85* cannot be cloned from WT lines, but can be cloned from OE transgenic lines. (**C**) Expression levels of *HpWRKY85* in rosette leaves of WT and five OE lines analyzed by RT-PCR. (**D**,**E**) The phenotype, primary root length, and lateral root numbers of WT and OE seedlings under normal condition (control) and D-mannitol. Each value represents the average of three replicates, and the error bars represent ±SD. Different letters indicate significant differences from the control (*p* < 0.05) tested by one-way ANOVA.

**Figure 6 ijms-24-00352-f006:**
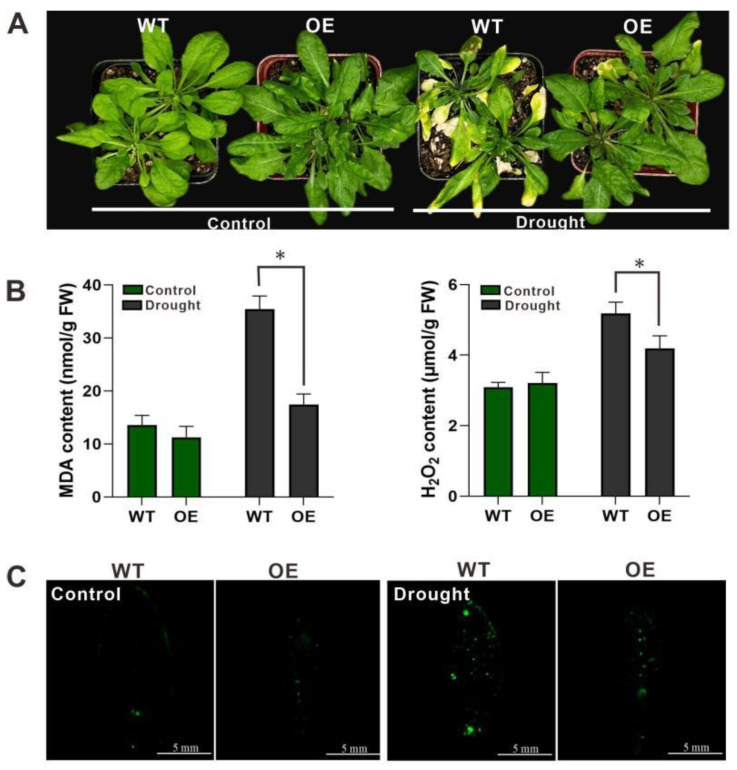
Drought tolerance test. (**A**) Four-week-old plants of WT and OE in soil were dehydrated for 8 days. (**B**) MDA and H_2_O_2_ accumulation in WT and OE lines under drought stress. Each value represents the average of three replicates, and the error bars represent ±SD. * indicate significant differences from the control (*p* < 0.05) tested by one-way ANOVA. (**C**) Histochemical staining with fluorescent CM-H_2_DCFDA to detect reactive oxygen species in WT and OE lines under drought stress.

## Data Availability

Not applicable.

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
