# Peer review of "Genome-Wide Identification of the Hypericum perforatum WRKY Gene Family Implicates HpWRKY85 in Drought Resistance"

_ijms, 2022, doi:10.3390/ijms24010352_

Round 1

Reviewer 1 Report

The manuscript “Genome-wide identification of the Hypericum perforatum WRKY gene family implicates HpWRKY85 in drought resistance” describes the identification and annotation of 86 WRKY candidate genes from the Hypericum perforatum genome. The WRKY candidate genes were divided into three groups based on phylogenetic analysis. The paper provided expression analysis of 10 WRKY candidate genes and functional characterization of the HpWRKY85, which showed to be involved in hormone induction and drought stress treatments. In addition, the HpWRKY85 protein exhibited nuclear subcellular localization, as expected for a transcription factor. Please, find below a list of recommendations for improvement of the manuscript.

Abstract

Line 23: “based on the whole genome database of H. perforatum”. What does it mean? I understood that authors have used the whole genome from H. perforatum to predict and identify WRKY candidate genes. However, classifying them into three groups was based on their sequence’s similarity (phylogenetic analysis). Wasn’t it?

Introduction

Lines 85-86: Please, see the comments above from line 23

Line 88: Change “Transcriptome” to RNA-seq analysis

Line 91: Change “verify” to validate

Line 92: Change “with different” to exhibit different

Line 93: “validate the expression position of HpWRKY”. How? Does not the subcellular expression assay use to identify where the HpWRKY protein is localized?

Line 94: What does mean “screened”? Did you mean functional characterization?

Results

Lines 126-134: AtWRKY72 uses…mechanism.” Does it make more sense to write these lines in the discussion section?

Lines 227-233: How were the times of the plant tissues collected for the RNA seq assay in figure 3A. Please, add this information to the legend. Explain in the legend the meaning of the x-axis (Fig. 3B).

Line 462: Add information of the times that plant tissues were collected (Fig. S4).

Lines 211-225 (Fig 3B): if the authors wish to show the difference in gene expression among treatments (SA, MeJA, and drought), I would suggest plotting the graphs by adding the inducing agents on the x-axis so that readers will be able to compare them. The time course could be transferred to Supplementary Materials.

Line 219: Take off “then”. It is repeated in the sentence.

Line 235: Add the information on why the HpWRKY85 was chosen to further analysis.

Line 237: Change “HpWRKY85” to HpWRKY85 as the authors are referring to the protein.

Line 238: After “nucleus” call the table where is described the information about the HpWRKY85 protein.

Line 240: “By transforming Arabidopsis”. Is it a stable or transient expression?

Lines 242-243: Change “were only expressed” to was exclusively expressed.

Line 244: Change the sentence “WRKY are known to bind to the W-box sequence (C/T)TGAC(T/C)” to Since WRKY transcript factors are known to bind to the W-box sequence (C/T)TGAC(T/C), our yeast-one hybrid test demonstrated that…

Line 254: Before “The seeds of Arabidopsis” add a sentence explaining what hypothesis is addressed with the overexpression of the HpWRKY85 assay.

Line 258: “HpWRKY85 were cloned from ten candidate transgenic lines”. Wasn’t it a PCR result?

Line 271: What plant tissue was used to set up semi-quantitative RT-PCR? Add this information to the result or legend in Fig. 5D.

Line 277: “field”. What does it mean? Arabidopsis plants cultivated in the field? If yes, what conditions?

Line 278-279: I found some words written incorrectly. Please double-check not only these lines but the whole manuscript.

Author Response

Response: 

Thank you so much for your affirmation of our work and the specific comments and suggestions. Here is my reply. Please review:

  1. Sorry for the misunderstanding, I have revised it.Please see Line 24-25.
  2. Please see Line 88-89.
  3. Please see Line 91 and 380.
  4. Thank you for your comments. Please see Line 94.
  5. Please see Line 95.
  6. Yes, subcellular expression assay was used to identify the location of HpWRKY protein. I have made changes. Please see Line 97.
  7. Please see Line 97.
  8. Thank you for your suggestion. AtWRKY72 and AtWRKY75 were mentioned here to predict the possible functions of homologous genes in perforatum. It is also a reasonable prediction of the results. So we decided to put this sentence into the result after considering.
  9. Thank you for your comments. Considering that RNA-seq related information has been published, this paper does not describe too much. Sampling information has been added to the legend of Fig 3A. Please see Line 237.

The meaning of the x-axis in Fig 3B has been specified in detail in legend. The relative expression is calculated using the generally accepted 2−ΔΔCt.

  1. Please see Line 481-482.
  2. Thank you for your comments. Fig 3B is not intended to show the differences of genes under different inducers or stresses, so we believe that the expression trend of each gene under different inducers is not comparable. In addition, our previous articles on the presentation of expression patterns are all like this. We think it is simple, clear and easy to understand.
  3. Sorry for the mistake. Please see Line 228.
  4. was exclusively expressed Please see Line 245-246.
  5. Sorry for the mistake. I have checked the full text and changed all protein names from italics to normal format.
  6. Please see Line 256.
  7. Using Arabidopsisprotoplasts to explore subcellular localization of genes is transient transformation.
  8. Thank you for your comments. Please see Line 255.
  9. Thank you for your comments. Please see Line 256-258.
  10. Thank you for your comments. Please see Line 267-268.
  11. Yes,it’s a PCR result. The legend of Fig 5B has also been clarified.
  12. Thank you for your comments. Please see Line 286.
  13. Field means soil culture. The culture conditions are still in the light incubator.
  14. Thank you so much for your affirmationand comments of our work. According to what you said, I have corrected all the spelling errors, grammar errors, inappropriate words and non-solid citations of literatures in this manuscript.

Reviewer 2 Report

Only minor edits in the attached document

Author Response

Thank you so much for your affirmation and comments of our work. According to what you said, I have revised it according to your comments. Please see Line 91-93, Line233-234, Line 318-320, Line 356. And the spelling errors, grammar errors, and inappropriate words in the text are corrected.
